# Indocyanine Green (ICG) Fluorescence-Assisted Open Surgery Using the Rubina^®^ Lens System in the Pediatric Population: A Single-Center Prospective Case Series

**DOI:** 10.3390/children11010054

**Published:** 2023-12-30

**Authors:** Ciro Esposito, Claudia Di Mento, Annalisa Chiodi, Mariapina Cerulo, Vincenzo Coppola, Fulvia Del Conte, Francesca Carraturo, Giovanni Esposito, Maria Escolino

**Affiliations:** 1Pediatric Surgery Unit, Department of Translational Medical Science, Federico II University, 80131 Naples, Italy; claudia.dimento@unina.it (C.D.M.); annalisa.chiodi@hotmail.it (A.C.); mariapina.cerulo@unina.it (M.C.); vincenzo.coppola2@unina.it (V.C.); fulviadelconte@gmail.com (F.D.C.); francesca.carraturo@unina.it (F.C.); x.escolino@libero.it (M.E.); 2CEINGE Biotechnology Center, 80131 Naples, Italy; pedsurg.esposito@unina.it

**Keywords:** ICG, fluorescence, NIRF, imaging, open surgery, pediatrics

## Abstract

Introduction: There are scarce papers about the use of fluorescence-guided surgery (FGS) in the open surgical field. This study aimed to assess the usefulness of FGS in an open setting in the pediatric population and to report our preliminary experience using the Rubina^®^ Lens system. Methods: All patients undergoing ICG fluorescence-assisted open surgery over the period September 2022–September 2023 were enrolled. Each surgical procedure was performed using the Rubina^®^ Lens for ICG fluorescence visualization. Results: A total of 25 patients, 14 boys and 11 girls with a median age at surgery of 5.8 years-old (range 0–15), were enrolled. Surgical indications were dermoid/epidermoid cysts of the head (*n* = 7), lymphangiomas of the head/neck (*n* = 2), thyroglossal duct cysts (*n* = 7), gynecomastia (*n* = 3), preauricular fistula (*n* = 2), second branchial cleft fistula (*n* = 1), fibrolipoma of the shoulder (*n* = 1) and myofibroma of the gluteal/perineal region (*n* = 2). In all procedures, an intralesional injection of 2.5 mg/mL ICG solution using a 30-gauge needle was administered. No adverse reactions to ICG occurred. Median operative time was 68.6 min (range 35–189). The visualization of ICG-NIRF with the Rubina^®^ Lens was achieved in all cases. No intraoperative complications were reported. Postoperative complications occurred in 3/25 patients (12%), with gynecomastia (*n* = 1), thyroglossal duct cyst (*n* = 1) and neck lymphangioma (*n* = 1), who developed a fluid collection in the surgical site, requiring needle aspiration in outpatient care (Clavien–Dindo 2). Complete mass excision was confirmed with pathology reports. Conclusions: Based on this initial experience, FGS using the Rubina^®^ Lens was very helpful in open surgery, providing enhanced visualization of anatomy and identification of margins, real-time reliability and low complication rate. It was easy to use, time saving, feasible and clinically safe. Previous experience in MIS is necessary to adopt this technology. The accuracy of the injection phase is important to avoid diffusion of the ICG into the perilesional tissue.

## 1. Introduction

Over the last few years, ICG fluorescence technology has become a valid support to pediatric surgeons in different indications in both open and minimally invasive surgery (MIS) [1,2,3]. This technology is nowadays largely used to improve surgical decision making intraoperatively, detect hidden masses and have an accurate visualization of the anatomic structures. Other advantages are present for visualization of tissue perfusion, vascularization and lymphatic flow [3]. Its use in pediatric MIS has been widely described in both laparoscopy and robotics [4,5,6,7,8]. Nevertheless, the open field, especially in the pediatric population, remains less explored [2]. One of the main indications in open surgery is for mass excision. Common congenital developmental masses include: thyroglossal duct cysts, branchial cleft cysts, dermoid cysts, vascular malformations, hemangiomas and lymphangiomas [9]. Other lesions are represented by common benign neoplastic lesions and include pilomatrixomas, lipomas, fibromas, neurofibromas and salivary gland tumors. Such pathologies have high incidence of recurrence, leading to hospital readmission and reoperation [10]. To prevent complications or recurrence, the surgery must be as accurate and precise as possible. Fluorescence-guided surgery (FGS) consists of a fluorescent dye, a light source (laser or LED to excite the dye at a specific wavelength) and a detection system such as a camera collecting the light emitted from the dye at a specific wavelength [11,12]. Indocyanine green (ICG) is one of the most used dyes in surgery thanks to its excellent safety profile and short half-life of 3–4 min. It undergoes no significant extrahepatic or enterohepatic circulation, and it is cleared exclusively through the liver. ICG is normally confined to the vascular stream, perfect for visualization of vascular perfusion, and entirely excreted into the biliary tract within a few hours from its injection, rendering it useful for visualizing biliary tract and flow [13,14]. The employment of fluorescence imaging has gained great popularity in many fields of adult surgery where it has demonstrated great potential in both open surgery and MIS [15,16]. In adults, ICG near-infrared fluorescence (NIRF) has been described in open surgery for sentinel lymph node biopsy in breast surgery and melanoma using a specifically designated camera [17,18]. Its use in the pediatric population, especially in open surgery, is still less explored. In the pediatric literature there are few case reports of detection and eradication of peritoneal metastases secondary to hepatoblastoma under ICG-NIRF imaging [19,20]. This study aimed to assess the usefulness of FGS in an open setting in the pediatric population and report our preliminary experience using the Rubina^®^ Lens system.

## 2. Materials and Methods

### 2.1. Patient Selection

All patients undergoing ICG fluorescence-guided open surgery over the period of September 2022–September 2023 were enrolled. The data were collected in a prospective manner. Exclusion criteria were patients with documented allergy or hypersensitivity to shellfish or iodides, thyroid disorders, and those who were using, at time of surgery, specific drugs, such as haloperidol, heparin, nifedipine, nitrofurantoin, phenobarbital, primidone and propranolol, which may modify the clearance of ICG. Informed consent to participate in this study was obtained from all participants or their parents or legal guardian in the case of children under 18 years of age. We disclose any financial relationships between the authors and the company.

### 2.2. Equipment

ICG fluorescence was obtained using the Rubina^®^ Lens system, manufactured by KARL STORZ SE & CO. KG, Tuttlingen, Germany. The holding arm system, VITOM^®^, manufactured by KARL STORZ SE & CO. KG, Tuttlingen, Germany, offers visualization possibilities for microsurgical and open surgical interventions. Technical specificities of the equipment are available at https://www.karlstorz.com/it/it/category.htm?cat=1000161024 (accessed on 28 November 2023). The vial of ICG (5 mg/mL) was reconstituted with 10 mL of sterile water to create a 2.5 mg/mL solution and injected using a 30-gauge needle. The volume of ICG solution that was injected was established according to the mass size, to avoid excessive increase in intralesional pressure and risk of lesion rupture. In lesions with a volume less than 5 mL, an intralesional injection of 0.5 to 2 mL ICG solution was administered. In large masses with a volume greater than 10 mL, up to 5 mL of ICG solution was injected.

### 2.3. Parameters Evaluated

Data about patient demographics, intraoperative details, and postoperative outcomes were assessed and included: age, gender, surgical indication, operative time (minutes), site of ICG administration, dosage of ICG, quality of intraoperative ICG-NIRF visualization, intraoperative complications and postoperative complications. The operative time was calculated after induction of general anesthesia, from the placement of sterile drapes until the completion of skin sutures. In ICG-NIRF group, the total operative time also included the entire laparoscopic tower assembly, optics assembly and ICG injection. Postoperative complications were graded according to the Clavien–Dindo classification [21]. To ease data interpretation and put it into perspective, we added a matched control group of patients on each type of intervention, to compare complications and duration of surgery.

### 2.4. Statistical Analysis

Statistical analysis was carried out using the Statistical Package for Social Sciences (SPSS Inc., Chicago, IL, USA), version 13.0. The demographic data were compared using the Student’s *t*-test. The categorical variables were compared using an χ^2^ test. Significance was defined as *p* value < 0.05.

## 3. Results

A total of 25 patients, who received ICG fluorescence-guided open surgical procedures during the study period, were enrolled. Patients included 14 boys and 11 girls, with a median age at surgery of 5.8 years-old (range 0–15). Surgical indications were dermoid/epidermoid cysts of the head (*n* = 7), lymphangiomas of the head/neck (*n* = 2), thyroglossal duct cysts (*n* = 7), gynecomastia (*n* = 3), preauricular fistulas (*n* = 2), a second branchial cleft fistula (*n* = 1), a fibrolipoma of the shoulder (*n* = 1) and myofibroma of the gluteal/perineal region (*n* = 2). Surgical excision of head, neck and thoracic masses were performed with minimal margins and primary closure of the defect using interrupted stitches in all patients. In all procedures, an intralesional injection of 2.5 mg/mL ICG solution using a 30-gauge needle was administered and ICG-NIRF imaging was obtained in real time beginning just a few seconds following the injection. No adverse reactions to ICG occurred. The visualization with the Rubina^®^ Lens always showed clear differentiation between pathologic and surrounding normal tissues. In all procedures, the NIRF signal lasted for the entire duration of the surgery. No intraoperative complications were reported. Postoperative complications occurred in 3/25 patients (12%), with gynecomastia (*n* = 1), thyroglossal duct cyst (*n* = 1) and neck lymphangioma (*n* = 1), who developed a fluid collection in the surgical site, requiring needle aspiration in outpatient care (Clavien–Dindo 2). Complete mass excision was confirmed with pathology reports. Surgical margins enlargement was not required in any patient. At a median follow-up of 11 months, no patients presented recurrence of the pathology or other complications related to the procedure.

Patient demographics and outcomes in the ICG-NIRF group and control group are reported in Table 1.

The surgical details of the different surgical procedures are reported below.

### 3.1. Dermoid/Epidermoid Cysts of the Head

These included supraorbital cysts (*n* = 4), scalp cysts (*n* = 2) and a supranasal cyst (*n* = 1). Patients included two boys and five girls with a median age at surgery of 2.3 years-old (range 0.8–12). Intralesional injections of 0.5 to 1 mL of ICG solution (2.5 mg/mL), using a 30-gauge needle, were administered during the operation according to the mass size. ICG-NIRF imaging provided clear visualization of cyst margins. In the case of a dermoid cyst of the skull, ICG-NIRF also allowed for the identification and ligation of the supplying vessel of the cyst, which originated from the cranial bone, without any bleeding. No complications or recurrence were reported at the median follow-up of 9 months (range 1–11).

The matched control group, without ICG-NIRF assistance, included supraorbital cysts (*n* = 6), a scalp cyst (*n* = 1) and a supranasal cyst (*n* = 1). Patients included four boys and four girls with a median age at surgery of 3.1 years-old (range 1–14). The comparative analysis showed significantly lower median operative time in the ICG group compared to the control group (21.6 vs. 37.5 min) [*p* = 0.001]. No complications occurred in either group.

### 3.2. Lymphangiomas of the Head/Neck

Two male patients, with a median age at surgery of 9.5 years-old (range 5–14), presented with a giant lymphangioma (>6 cm) in the nucal and submandibular region, respectively. Indication for surgical removal instead of ultrasound-guided sclerosis was decided, based on the large size of the malformation and its close position to main vessels. In such cases, 3 to 5 mL of ICG solution (2.5 mg/mL) were injected into the lesion, allowing for real-time identification of the mass corners and ramifications (Figure 1). Complete resection of the mass was visually confirmed as fluorescence was not yet visible in the surrounding normal tissue. A subcutaneous drain was left in both patients for up to 48 h postoperatively. Pathology confirmed diagnosis of cystic lymphangioma and negative resection margins in both patients. The patient with lymphangioma of the nucal region developed fluid collection in the surgical site 3 weeks following surgery. US imaging showed the fluid collection in the surgical site and no residual lymphatic malformation. The patient underwent needle aspiration in the outpatient care (Clavien–Dindo 2), with resolution of the collection and no further recurrence. At a median follow-up of 7.5 months (range 2–10), no disease recurrence was observed.

The matched control group, without ICG-NIRF assistance, included two girls with a lymphangioma of the lateral region of the neck. The median patient age was 7.3 years-old (range 3.5–14). One patient developed postoperative fluid collection in the surgical site, which was managed with needle aspiration (Clavien–Dindo 2). The comparative analysis showed significantly lower median operative time in the ICG group compared to the control group (33.9 vs. 49 min) [*p* = 0.001]. The complications rate was not different between the two groups (1/2, 50% vs. 1/2, 50%) [*p* = 0.88].

### 3.3. Thyroglossal Duct Cysts

Seven patients with thyroglossal duct cysts were operated on using ICG-NIRF. There were four girls and three boys, with a median age at surgery of 4.7 years-old (range 2.5–9). Indication for surgery was a rapid increase in size (*n* = 4) and infection (*n* = 3). Infected cysts were operated after resolution of the acute episode. When an external fistulous orifice connected to the cyst was clearly identifiable on the median anterior region of the neck, it was cannulated using an abocath catheter and 0.5 to 2 mL of ICG solution (2.5 mg/mL) were injected through it which allowed clear identification of the fistula tract, the cyst and the portion of it that was adherent to the body of hyoid bone, which was excised together with the cyst without any bleeding (Figure 2). In patients with an isolated cyst without external fistulisation, 1 to 2 mL of ICG solution were directly injected into the cyst, after isolating it, allowing for clear demarcation of the cyst dome and its origin from the hyoid bone. A laminar drain was left in all cases for at least 12–24 h postoperatively. Only one patient developed fluid collection in the surgical site 2 weeks following surgery and underwent needle aspiration in the outpatient (Clavien–Dindo 2), with resolution of the collection with no further recurrence. At a median follow-up of 8.5 months (range 1–13), no disease recurrence was observed.

The matched control group, without ICG-NIRF assistance, included 10 patients. There were four boys and six girls with a median age at surgery of 5.5 years-old (range 3–11). One patient developed postoperative surgical site infection, treated with antibiotic therapy (Clavien–Dindo 2). The comparative analysis showed a significantly lower median operative time in the ICG group compared to the control group (64.5 vs. 95 min) [*p* = 0.001]. The complications rate was not different between the two groups (1/7, 14.3% vs. 1/10, 10%) [*p* = 0.66].

### 3.4. Gynecomastia

Three boys with bilateral gynecomastia grade 3, according to Simon classification, underwent subcutaneous mastectomy through sub-areolar access associated with liposuction. The median patient’s age at surgery was 13.7 years-old (range 11–15). After the induction of anesthesia, 2 mL of ICG solution (2.5 mg/mL) was administered intradermally into the periareolar region at multiple points in the four quadrants of the breast. Light massage of the breast for 2 to 3 min was undertaken to facilitate ICG mobility into the mammary gland. Subsequently, real-time ICG-NIRF observations allowed for the identification of the mammary gland margins and to leave at least 5–10 mm of gland thickness below the nipple–areola complex (NAC). Finally, ICG-NIRF was helpful for checking perfusion of NAC following resection of the gland to avoid ischemic injury and prevent nipple retraction or atrophy postoperatively (Figure 3). A suction drain tube was placed in each cavity at the end of the procedure and removed when the output volume was less than 40 mL/day. One patient developed unilateral seroma in the breast region, requiring needle aspiration for 2 weeks in outpatient care (Clavien–Dindo 2), with resolution of the collection and no further recurrence. At a median follow-up of 7.5 months (range 1–10), no disease recurrence was observed.

The matched control group, without ICG-NIRF assistance, included nine boys with bilateral gynecomastia. The median patient’s age at surgery was 14.6 years-old (range 12–16). Five out of the nine patients (55.5%) developed postoperative seroma in the breast region, managed with needle aspiration in an outpatient setting (Clavien–Dindo 2). The comparative analysis showed significantly lower median operative time in the ICG group compared to the control group (137.5 vs. 189 min) [*p* = 0.001]. The complications rate was significantly higher in the control group (5/9, 55.5%) compared to the ICG group (1/3, 33.4%) [*p* = 0.001].

### 3.5. Preauricular Fistula and Second Branchial Cleft Fistula

Two patients (a 13-year-old boy and a 12-year-old girl) with a preauricular fistula and a 3-year-old girl with a second branchial cleft fistula, underwent ICG-NIRF-assisted resection. In all patients, the external fistulous orifice connected to the underlying cyst was clearly identified and cannulated using an abocath catheter. About 1 mg of ICG solution (2.5 mg/mL) was injected through the catheter and allowed for the clear identification of the fistula tract and the underlying cyst, which were excised as whole without any bleeding. No drains were left at the end of the procedure. The postoperative course was uneventful, without any recurrence or complication at the median follow-up of 11 months (range 8–12).

The matched control group, without ICG-NIRF assistance, included five boys with a preauricular fistula. The median patient’s age at surgery was 9.8 years-old (range 6–14). Two out of the five patients (40%) developed postoperative complications, including surgical site infection and serous discharge from the surgical site, both managed with antibiotic therapy (Clavien–Dindo 2). The comparative analysis showed significantly lower median operative time in the ICG group compared to the control group (44.9 vs. 76.7 min) [*p* = 0.001]. The complications rate was significantly higher in the control group (2/5, 40%) compared to the ICG group (0) [*p* = 0.001].

### 3.6. Other Applications

A 15-year-old boy with fibrolipoma of the shoulder and two boys, aged 2- and 6- years-old, with myofibroma of the gluteal and perineal region, received ICG-NIRF-assisted resection. During the procedure, 2 to 3 mL of ICG solution (2.5 mg/mL) were injected into the lesion which guided in real-time the mass resection, allowing for the visualization of the resection margins and the vessels supplying the mass (Figure 4). No drains were left at the end of the procedure. The postoperative course was uneventful, without any recurrence or complication at the median follow-up of 8 months (range 2–12).

## 4. Discussion

This study enrolled 25 patients undergoing resection of head, neck or trunk masses using ICG-NIRF assistance. Our goal was to demonstrate that ICG-NIRF assistance can be a feasible and reliable method to assess mass anatomy intraoperatively, reduce the risk of complications and avoid reoperations for recurrence due to poor visualization of margins, anatomical variants and adjacent structures’ involvement. We adopted the Rubina^®^ Lens system for ICG-NIRF imaging to visualize resection margins and perform video-assisted complete excision awaiting pathological confirmation. In our experience, FGS demonstrated to be a valid support to achieve radical excision of masses, especially when the anatomy is not clear, making it difficult to differentiate the mass and its corners from the surrounding tissues. The comparative analysis with matched control groups of patients showed that ICG-NIRF assistance was associated with significantly lower duration of surgery in each type of intervention. The complications rate was decreased with ICG-NIRF assistance only for gynecomastia and preauricular fistulas. In such indications, ICG-NIRF was helpful to decrease postoperative morbidity and avoid additional procedures to manage the complications that occurred.

There is very scarce pediatric literature about use of ICG fluorescence in open surgery. The main applications have been described for oncological indications to facilitate lymph node sampling in Wilms’ tumors, aid in hepatectomy for hepatoblastoma or localize pulmonary metastases of pediatric solid tumors [22,23,24,25]. Different studies have examined the feasibility of ICG-guided tumor resection in common childhood solid tumors such as neuroblastoma, sarcomas, hepatic tumors, pulmonary metastases and other rare tumors [5,24,25]. Fluorescence-guided lymphatic mapping of Wilms tumor drainage is feasible by both intraparenchymal and peri-hilar injection techniques. However, whether lymphatic mapping improves the precision of lymph node sampling is unknown and should be studied in prospective trials [22,23]. Two recent systematic reviews assessed that, although the number of available studies is small, ICG-guided angiography might be useful for intraoperative intestinal perfusion assessment and other indications, perhaps even more than conventional clinical assessment [8,9]. However, the quality of evidence supporting such applications currently appears low [9].

We first described ICG-NIRF-assisted resection of dermoid/epidermoid cysts, thyroglossal duct cysts, gynecomastia and lymphangioma of the head/neck. No other reports about these indications are available in the existing pediatric literature. The main advantages of using this technology in such frequent indications were to fasten surgery, avoid intra and postoperative complications, improve the learning curve, and address the shortcomings and disadvantages.

In open surgery as well as in laparoscopy, specific equipment for ICG-NIRF visualization is needed. In this series, the Rubina^®^ Lens system by KARL STORZ was adopted in all procedures (https://www.karlstorz.com/it/it/category.htm?cat=1000161024, accessed on 28 November 2023).

ICG-NIRF technology provides several advantages that make it affordable for daily use in the OR. First, it is easy to use; the ICG solution can be rapidly prepared, and administration only requires intralesional injection. The amount of ICG solution to inject can be established in real time in accordance with the volume of the mass. In our experience, a maximum injection volume of 5 mL was adopted for giant lymphangiomas of the head and neck. The assembly of equipment needed for ICG-NIRF (laparoscopic tower, optics and holding arm) in the OR may be time-consuming, but only at the beginning of the experience. If the OR team is well trained, the preoperative preparation will not cause any delay of the surgical operation. After the first cases, it required no more than 5 min for our trained team. As reported in our series, despite the additional steps needed for equipment assembly, the surgery in ICG-NIRF groups was always significantly faster compared with control groups for all types of surgical procedures analyzed. This technology is useful for better anatomic delineation and image-assisted radical resection of difficult masses. Pathology reports confirmed complete mass excision in all patients.

One of the main criticisms of using this technology in open surgery is that the intralesional injection of ICG in a dermoid/epidermoid cyst and much more so in a thyroglossal duct cyst implies disruption of the capsule, leading to potentially increased risk of recurrence. To obviate such risks, we use a 30-gauge needle that creates a very narrow hole in the capsule thus avoiding spreading of the lesion’s content. Additionally, the volume of the ICG solution injected was established according to the mass size, to avoid excessive increase in intralesional pressure and the risk of lesion rupture. In lesions with a volume less than 5 mL, intralesional injection of 0.5 to 2 mL ICG solution were administered. In large masses with a volume greater than 10 mL, up to 5 mL of ICG solution were injected. In thyroglossal cysts with a cutaneous fistula, we preferentially injected the ICG solution through the connecting fistula and we obtained clear visualization of the cyst and its attachment to the hyoid bone. Following the intralesional injection, the distribution of the dye may not allow for the clear visualization of all corners and ramifications of the lesion itself. Obviously, this depends on several factors, such as the lesion’s size and depth, vascularization and inflammation. To obviate such drawbacks, in the case of giant masses with incomplete labelling, we repeated intralesional injection in a different site to achieve better visualization of the lesion’s corners and/or ramifications. We removed any “labeled” tissue to ensure radical excision and we did not observe distribution of the dye to normal tissues. It is safe for the OR team as there is no exposure to radiation intraoperatively, as well as for the patients, who did not report any adverse reaction to ICG administration. Three patients from our series developed postoperative seroma formation in the surgical site, but none of these was directly attributable to ICG injection or use of ICG-NIRF. Finally, it is cheap as it only requires the costs of ICG dye (about 40 euros/vial) whereas the technical equipment for ICG-NIRF visualization is reusable. Additionally, considering that these types of skin surgeries such as dermoid cysts or thyroglossal cysts are usually performed by young residents, the use of ICG-NIRF assistance should be seen as an adjunctive learning tool to provide better visualization of anatomical details, improve eye-hand coordination and flatten the learning curve.

## 5. Conclusions

Based on this initial experience, FGS using the Rubina^®^ Lens was very helpful in open surgery, providing enhanced visualization of anatomy and identification of margins, real-time reliability and a low complication rate. It was easy to use, time saving, feasible and clinically safe. Previous experience in MIS is necessary to adopt this technology. The accuracy of the injection phase is important in order to avoid diffusion of the ICG into the perilesional tissue. Further prospective series, with a larger number of patients and a longer follow-up period, are expected to validate these preliminary results and expand applications and indications of this promising technology.

## Figures and Tables

**Figure 1 children-11-00054-f001:**
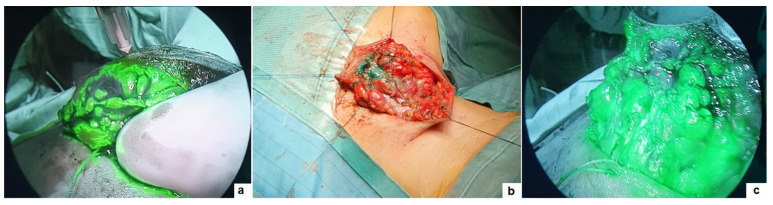
Lymphangioma of the submandibular region: intralesional injection of ICG (**a**); visualization of the mass on white light (**b**) and ICG-NIRF (**c**).

**Figure 2 children-11-00054-f002:**
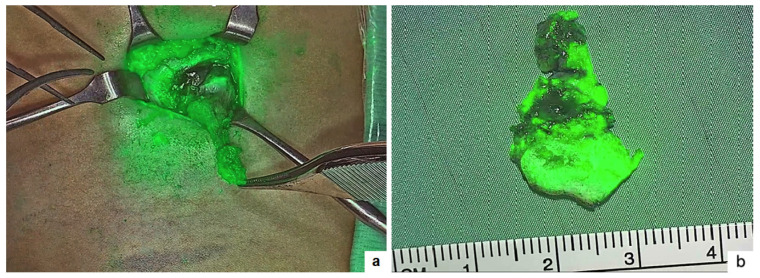
Thyroglossal duct cyst: isolation of the fistula and underlying cyst on ICG-NIRF (**a**) and resection of the cyst together with the body of hyoid bone (**b**).

**Figure 3 children-11-00054-f003:**
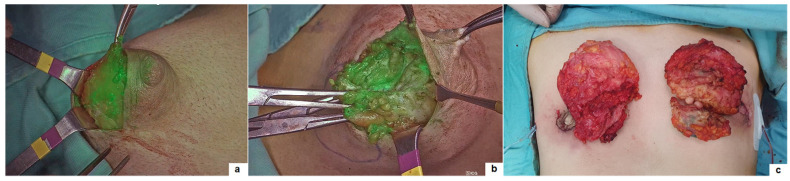
Gynecomastia: identification on ICG-NIRF of the mammary gland margins (**a**); perfusion of the nipple–areola complex (**b**) and the resected specimens (**c**).

**Figure 4 children-11-00054-f004:**
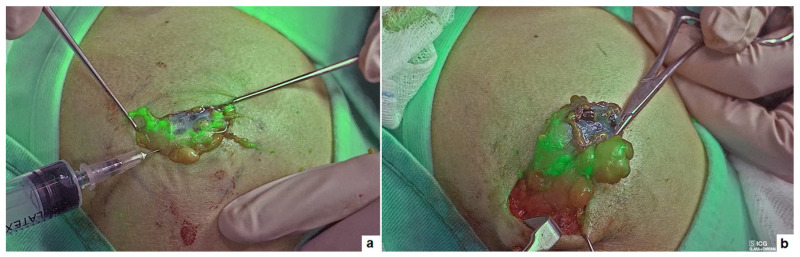
Fibrolipoma of the shoulder: intralesional injection of ICG (**a**) and visualization of the resection margins on ICG-NIRF (**b**).

**Table 1 children-11-00054-t001:** Patients’ demographics and outcomes in the ICG-NIRF group and control group.

Parameter	ICG-NIRF Group (*n* = 25)	Control Group (*n* = 34)
M/F, *n*/*n*	14/11	22/12
Median patient age, years (range)	5.8 (0–15)	8.0 (1–16)
Surgical indications, type (*n* =)	Dermoid/epidermoid cysts of the head (7) Lymphangiomas of the head/neck (2) Thyroglossal duct cysts (7) Gynecomastia (3) Preauricular fistula (2) Second branchial cleft fistula (1) Fibrolipoma of the shoulder (1) Myofibroma of the gluteal/perineal region (2)	Dermoid/epidermoid cysts of the head (8) Lymphangiomas of the head/neck (2) Thyroglossal duct cysts (10) Gynecomastia (9) Preauricular fistula (5)
Intraoperative complications, *n* = and type (%)	0	0
Postoperative complications, *n* = and type (%)	3/25 seroma (12%)	7/34 seroma (20.5%) 2/34 surgical site infection (5.8%)
Recurrence, *n* (%)	0	0

## Data Availability

The data presented in this study are available on request from the corresponding author, C.E. The data are not publicly available due to privacy or ethical restrictions.

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
