# Peer review of "Indocyanine Green (ICG) Fluorescence-Assisted Open Surgery Using the Rubina® Lens System in the Pediatric Population: A Single-Center Prospective Case Series"

_children, 2023, doi:10.3390/children11010054_

Round 1

Reviewer 1 Report (Previous Reviewer 1)

Comments and Suggestions for Authors

According to the authors, the main advantages of using this technology in such frequent indications were to fasten surgery, avoid intra- and post-operative complications, improve the learning curve.

However, the entire laparoscopic tower assembly, optics assembly and ICG injection delay the surgery, not speed it up. If the authors claim this, they should prove it with a comparative study. 

Similarly, the authors claim that the use of ICG in this type of surgery decreases intraoperative and postoperative complications, which are already minimal in the surgeries described in the article. Again, a comparative design is needed to conclude this.

It is difficult to argue that ICG in this type of surgery improves learning curve, because these types of skin surgeries such as dermoid cysts or thyroglossal cysts are usually performed by first or second year residents. The use of optics and laparoscopic screens in this type of surgery makes the learning curve for open surgery more difficult.

Comments on the Quality of English Language

According to the authors, the main advantages of using this technology in such frequent indications were to fasten surgery, avoid intra- and post-operative complications, improve the learning curve.

However, the entire laparoscopic tower assembly, optics assembly and ICG injection delay the surgery, not speed it up. If the authors claim this, they should prove it with a comparative study. 

Similarly, the authors claim that the use of ICG in this type of surgery decreases intraoperative and postoperative complications, which are already minimal in the surgeries described in the article. Again, a comparative design is needed to conclude this.

It is difficult to argue that ICG in this type of surgery improves learning curve, because these types of skin surgeries such as dermoid cysts or thyroglossal cysts are usually performed by first or second year residents. The use of optics and laparoscopic screens in this type of surgery makes the learning curve for open surgery more difficult.

Author Response

Reviewer 1:

1. We already compared the operative times of each type of surgical procedure performed with or without ICG-NIRF assistance. The operative time was calculated after induction of general anesthesia, from the placement of sterile drapes till to the closure of skin incisions. In ICG-NIRF group, the total operative time also included the entire laparoscopic tower assembly, optics assembly and ICG injection, that, for our trained team, required no more than 5 minutes. We agree with the Reviewer that the assembly of the equipment necessary for ICG-NIRF may be time-consuming, but just at beginning of experience. If the OR team is well trained, the pre-operative preparation will not cause any delay of the surgical operation. As reported in our series, despite the additional steps needed for equipment assembly, the surgery in ICG-NIRF groups was always significantly faster compared with control groups for all types of surgical procedures analyzed.

We clarified these aspects in the Materials and Methods and Discussion sections, according to the Reviewer’s comment.

2. We agree with the Reviewer that the type of surgeries analyzed have minimal risk of intra- and post-operative complications. Nevertheless, our comparative analysis already showed that the complications rate was decreased by ICG-NIRF assistance for gynecomastia and pre-auricular fistula. In such indications, ICG-NIRF was helpful to decrease post-operative morbidity and avoid additional procedures to manage the complications occurred.

We added these considerations to the Discussion section, according to the Reviewer’s comment.

3. We agree with the Reviewer that these types of skin surgeries such as dermoid cysts or thyroglossal cysts are usually performed by young residents. In such optic, the use of ICG-NIRF assistance should be seen as an adjunctive learning tool to provide better visualization of anatomical details, improve eye-hand coordination and flatten the learning curve.

We added these considerations to the Discussion section, according to the Reviewer’s comment.

Reviewer 2 Report (Previous Reviewer 4)

Comments and Suggestions for Authors

The revision has substantially improved the manuscript. 

Comments on the Quality of English Language

Acceptable.

Author Response

  1. We edited the English language of manuscript, according to the Reviewer’s comment.

Reviewer 3 Report (Previous Reviewer 2)

Comments and Suggestions for Authors

Thank you for the revised manuscript, which has improved considerably and answered my questions sufficiently. No further comments. 

Author Response

  1. We thank the Reviewer for the comments.

Reviewer 4 Report (Previous Reviewer 3)

Comments and Suggestions for Authors

Dear authors,

Thank you for offering me the opportunity to review your paper. I really appreciate the adaptations made compared to the previous version. I have one minor suggestion to consider that would make the paper stronger in my opinion. I would add a table comparing the baseline data and demographics between the cases and the matched control group to make the data and the impact of your technique easier to grasp for readers.

Author Response

1. Following the Reviewer’s suggestion, we edited Table 1 by reporting patient demographics and outcomes in ICG-NIRF group and Control group.

Reviewer 5 Report (New Reviewer)

Comments and Suggestions for Authors

see attached file.

Comments on the Quality of English Language

see attached file.

Author Response

Reviewer 5:

1. Following the Reviewer’s suggestion, we provided a more detailed analysis of references regarding use of ICG-NIRF in the pediatric surgery field and their contribution to the study in the Discussion.

2. The vial of ICG (5 mg/mL) was reconstituted with 10 mL of sterile water to create a 2.5 mg/mL solution and injected using a 30-gauge needle. The volume of ICG solution to inject was established according to the mass size, to avoid excessive increase of intralesional pressure and risk of lesion rupture. In lesions with volume lower than 5 mL, intralesional injection of 0.5 to 2 mL ICG solution was performed. In large masses with volume higher than 10 mL, up to 5 mL of ICG solution were injected.

We added this information to the Materials and Methods and Discussion sections, according to the Reviewer’s question.

3. The manuscript has been reviewed by a native English speaker to improve language, according to the Reviewer’s comment. All editions in the text have been highlighted in red color.

Round 2

Reviewer 1 Report (Previous Reviewer 1)

Comments and Suggestions for Authors

The authors have responded adequately to the reviewers' questions.

Comments on the Quality of English Language

English language fine. No issues detected

This manuscript is a resubmission of an earlier submission. The following is a list of the peer review reports and author responses from that submission.

Round 1

Reviewer 1 Report

Comments and Suggestions for Authors

The authors present their series of open surgery guided by indocyanine green fluorescence. This type of pigment has been of great relevance in abdominal, thoracic and urological surgery, and they now present its indications in skin surgery. The manuscript is well structured and written, although there are several issues regarding its purpose and practical application.

The main impediment to the publication of this article is the clinical relevance of this type of procedure. On the one hand, the preparation of the ICG optics involves an increase in operative time, in addition to having a laparoscopic tower for a procedure that is superficial. This may represent a conflict of interest with other types of interventions in which the presence of the laparoscopic tower is essential, when in the procedures described by the authors, the use of Rubina optics is absolutely dispensable.

The authors should reconsider the publication of this article, because although the importance and usefulness of ICG in urological, abdominal and thoracic surgery is evident (as has been published in different types of studies), the application of ICG in all types of interventions may not be entirely indicated or justified.

These types of soft tissue tumour surgeries (dermoid cysts, thyroglossal duct cysts, etc.) are operations in which it is essential to preserve the capsule of the lesion to avoid accidental rupture and leakage of material into adjacent tissues.By injecting ICG intralesionally, the capsule of the lesion is being ruptured provocatively, which goes against the surgical principles of this type of intervention. ICG fluorescence has its limitations, it is not indicated for "all types of interventions", and "publication for publication's sake" should be avoided.If we do not break this trend, we could start operating on phimosis and ankyloglossia with ICG fluorescence.

Comments on the Quality of English Language

The authors present their series of open surgery guided by indocyanine green fluorescence. This type of pigment has been of great relevance in abdominal, thoracic and urological surgery, and they now present its indications in skin surgery. The manuscript is well structured and written, although there are several issues regarding its purpose and practical application.

The main impediment to the publication of this article is the clinical relevance of this type of procedure. On the one hand, the preparation of the ICG optics involves an increase in operative time, in addition to having a laparoscopic tower for a procedure that is superficial. This may represent a conflict of interest with other types of interventions in which the presence of the laparoscopic tower is essential, when in the procedures described by the authors, the use of Rubina optics is absolutely dispensable.

The authors should reconsider the publication of this article, because although the importance and usefulness of ICG in urological, abdominal and thoracic surgery is evident (as has been published in different types of studies), the application of ICG in all types of interventions may not be entirely indicated or justified.

These types of soft tissue tumour surgeries (dermoid cysts, thyroglossal duct cysts, etc.) are operations in which it is essential to preserve the capsule of the lesion to avoid accidental rupture and leakage of material into adjacent tissues.By injecting ICG intralesionally, the capsule of the lesion is being ruptured provocatively, which goes against the surgical principles of this type of intervention. ICG fluorescence has its limitations, it is not indicated for "all types of interventions", and "publication for publication's sake" should be avoided.If we do not break this trend, we could start operating on phimosis and ankyloglossia with ICG fluorescence.

Author Response

Reviewer 1:

1. There is very scanty pediatric literature about use of ICG fluorescence in open surgery. The main applications have been described for oncological indications to facilitate lymph node sampling in Wilms’ tumor, aid in hepatectomy for hepatoblastoma or localize pulmonary metastases of pediatric solid tumors (references 22-25). We first described ICG-NIRF assisted resection of dermoid/epidermoid cysts, thyroglossal duct cysts, gynecomastia, lymphangioma of head/neck. No other reports about these indications are available in the existing pediatric literature. The main advantages of using this technology in such frequent indications were to fasten surgery, avoid intra- and post-operative complications, improve the learning curve, and address the shortcomings and disadvantages. We agree with the Reviewer that one of the main criticisms of using this technology in open surgery is that the intralesional injection of ICG in a dermoid/epidermoid cyst and much more in a thyroglossal duct cyst implies disruption of the capsule, leading to potentially increased risk of recurrence. To obviate such risks, we commonly use a 30-gauge needle, that creates very narrow hole in the capsule, thus avoiding spreading of the lesion’s content. In cases of thyroglossal duct cyst with cutaneous fistula, we preferentially injected the ICG solution through the connecting fistula and we obtained the clear visualization of the cyst and its attachment to the hyoid bone. We added these considerations to the Discussion section and updated the References list, according to the Reviewer’s suggestion.

Reviewer 2 Report

Comments and Suggestions for Authors

The manuscript describes the use of intralesional injection of indocyanine green into different congenital benign malformations in 25 cases to secure complete resection of the mases. The technique and results in is well described and documented by illustrative figures. The manuscript is well-written, but the scientific merit of the paper is limited because of the diversity in pathologies and the few patients. This is also sufficiently pointed out by the authors. 

Mig biggest concern I how one can rely on that the dye is even distributed to all corners and ramifications of a given mass, e.g. a lipoma. This must depend on several factors. I am not convinced, that it can secure total/radical excision in all cases, but I do understand that any “labeled” tissue must be removed or is there a risk of distribution of the dye to normal tissue? 

Author Response

Reviewer 2:

1. We agree with the Reviewer’s comment that following the intralesional injection, the distribution of the dye may not allow the clear visualization of all corners and ramifications of the mass itself. Obviously, it depends on several factors, such as lesion’s size and depth, vascularization, inflammation. To obviate such drawbacks, in case of giant masses with incomplete labelling, we repeated intralesional injection in a different site to achieve better visualization of the mass corners and/or ramifications. We removed any “labeled” tissue to ensure radical excision and we did not observe distribution of the dye to normal tissues. We discussed these aspects in the Discussion section, according to the Reviewer’s comment.

Reviewer 3 Report

Comments and Suggestions for Authors

Dear authors,

Thank you for the opportunity to read your manuscript and provide feedback. I found it an interesting read. Please find my recommendations to strengthen your paper below:

- The introduction section provides little background information on the actual benefits and applications of fluorecense-guided surgery in adults and/or children. However, the first two paragraphs of the discussion section cover this. I would recommend moving those paragraphs here.

- The methods section and discussion section contain a lot of information about a specific type of instrument used for the surgery performed. I believe these sections can be reduced to 1-2 sentenced providing necessary information about the equipment for the reader, while all the equipment's technical specificities can be reduced to a link to the website if the reader would like to read more.

-Please clarify if the data was collected in a retrospective or prospective manner

- Given that the article explicitly states which brand of material they use, it is important to clarify in the methods section what the relationship between the authors and the company is and if the company had any financial interests in this study. I understand that part of this is covered in the acknowledgements section at the end of the article, however, in order to provide true transparency to the reader, I recommend adding it in the methods section as well.

- The results are interesting to read, but are very difficult to interpret and put into perspective without a comparisson group. I would recommend the authors to add a matched control group of patients, to compare complications and duration of intervention. The matching should be done on type of intervention at least, and if possible on age/gender/severity/etc to lower the rate of confounding.

- Please add a few paragraphs comparing and contrasting your data to existing literature in the discussion section.

Author Response

Reviewer 3:

1. We moved the first two paragraphs of the Discussion to the Introduction section, following the Reviewer’s suggestion. We edited the References list, accordingly.

2. We edited the Materials and Methods and Discussion sections, by reducing the information about the equipment’s technical specificities. We added a link to the website, as suggested by the Reviewer.

3. The data was collected in a prospective manner. We clarified this detail in the Materials and Methods section, according to the Reviewer’s suggestion.

4. We added financial disclosure statement to the Materials and Methods section, according to the Reviewer’s suggestion.

5. Following the Reviewer’s suggestion, we added a matched control group of patients on each type of intervention, to compare complications and duration of surgery, to ease the data interpretation and put it into perspective. The comparative analysis showed that ICG-NIRF assistance was associated with significantly lower duration of surgery in each type of intervention. The complications rate was decreased by ICG-NIRF assistance only for gynecomastia and pre-auricular fistula. We added this data to the Results and Discussion sections.

6. There is very scanty pediatric literature about use of ICG fluorescence in open surgery. The main applications have been described for oncological indications to facilitate lymph node sampling in Wilms’ tumor, aid in hepatectomy for hepatoblastoma or localize pulmonary metastases of pediatric solid tumors (references 22-25). We first described ICG-NIRF assisted resection of dermoid/epidermoid cysts, thyroglossal duct cysts, gynecomastia, lymphangioma of head/neck. No other reports about these indications are available in the existing pediatric literature. The main advantages of using this technology in such frequent indications were to fasten surgery, avoid intra- and post-operative complications, improve the learning curve, and address the shortcomings and disadvantages. We added these considerations to the Discussion section and updated the References list, according to the Reviewer’s suggestion.

Reviewer 4 Report

Comments and Suggestions for Authors

The title of the article is questionable. This is not a review or an expert panel, this is a low sample size, single-center retrospective case series. I don't think the authors can speak, with the data present, as to what the indications for the technique are.

The text needs English revision by a native speaker. Inappropriate verb tenses, unnatural expressions and incorrect terms ("sieroma") are used.

I do not think that the mean surgical time makes sense when procedures as disparate as an epidermoid cyst and a thyroglossal duct cyst are included.

Although I see the usefulness in cases such as gynecomastia, I find questionable the intralesional injection of ICG in a dermoid/epidermoid cyst (and much more in a thyroglossal duct cyst), which implies a disruption of the capsule and an increased risk of recurrences, without contributing anything with respect to conventional surgery. I recommend commenting on these aspects in the discussion

The scale of complications reported by the authors is Clavien-Dindo, not Clavien (as named in the abstract). It would be interesting to detail in the abstract to which patients this complication refers, since a collection in a dermoid is not the same as in a thyroglossal duct cyst.

It seems very debatable to me that this technique would decrease the surgical time without presenting a series of adequately matched controls.

Methods

a.       The study received the appropriate Institute Review Board (IRB) approval.” Provide approval code and date in the main text, not only on the last page.

b.       The Helsinki update to be used is the 2013 update, not the 2008 update.

c.       Figure 1 is not very illustrative. The parts of the system are not referenced, and multiple recognizable individuals are shown.

d.       It is not made explicit in the main text whether the work is prospective or retrospective.

Results

a.       Multiple data are missing, such as follow-up time or characteristics by pathology group.

b.       How do the authors know that the collection of the patient with lymphangioma was a seroma and not lymphorrhea? Why did authors perform a surgical excision and not an ultrasound-guided sclerosis?

c.        Figure 3. Image b illustrates a standard Sistrünk technique. Delete

d.       The cysts were rapidly increasing in size due to hemorrhage or infection and were operated after resolution of acute episode”. All 7?

e.       "Data supporting reported results are archived in the first author's personal datasets.". The reader is not interested in where the data are stored, the reader is interested in whether data are in the public domain or how he can get them if he needs them.

Comments on the Quality of English Language

.

Author Response

Reviewer 4:

1. We edited the Title of the manuscript as “Indocyanine Green (ICG) Fluorescence-Assisted Open Surgery Using Rubina® Lens System In The Pediatric Population: A Single-Center Prospective Case Series.”, according to the Reviewer’s suggestion.

2. The manuscript has been edited by a native English speaker, according to the Reviewer’s comment. All editions in the text have been highlighted in red color.

3. We added a matched control group of patients on each type of intervention, to compare complications and duration of surgery, to ease the data interpretation and put it into perspective. The comparative analysis showed that ICG-NIRF assistance was associated with significantly lower duration of surgery in each type of intervention. We added this information to the Results and Discussion sections, according to the Reviewer’s suggestion.

4. We agree with the Reviewer that one of the main criticisms of using this technology in open surgery is that the intralesional injection of ICG in a dermoid/epidermoid cyst and much more in a thyroglossal duct cyst implies disruption of the capsule, leading to potentially increased risk of recurrence. To obviate such risks, we commonly use a 30-gauge needle, that creates very narrow hole in the capsule, thus avoiding spreading of the lesion’s content. In cases of thyroglossal duct cyst with cutaneous fistula, we preferentially injected the ICG solution through the connecting fistula and we obtained the clear visualization of the cyst and its attachment to the hyoid bone. We added these statements to the Discussion section, according to the Reviewer’s comment.

5. We correctly renamed the Clavien-Dindo grading scale for complications throughout the manuscript. We also detailed in the Abstract to which patients the reported complications refer.

6. Following the Reviewer’s suggestion, we added a matched control group of patients on each type of intervention, to compare complications and duration of surgery, to ease the data interpretation and put it into perspective. The comparative analysis showed that ICG-NIRF assistance was associated with significantly lower duration of surgery in each type of intervention. The complications rate was decreased by ICG-NIRF assistance only for gynecomastia and pre-auricular fistula. We added this information to the Results and Discussion sections, according to the Reviewer’s suggestion.

Methods

a. We detailed IRB approval code and date in the Materials and Methods section, according to the Reviewer’s question.

b. We edited the Institutional Review Board Statement as follows: “Data were retrospectively evaluated according to the principles of the Declaration of Helsinki as revised in 2013.”, according to the Reviewer’s suggestion.

c. We removed Figure 1, following the Reviewer’s comment.

d. The data was collected in a prospective manner. We clarified this detail in the Materials and Methods section, according to the Reviewer’s suggestion.

Results

a. We reported patient characteristics and follow-up time by pathology group in the Results section, according to the Reviewer’s question.

b. We defined the post-operative collection of the patient with lymphangioma as seroma and not lymphorrhea, because post-operative US imaging showed the fluid collection in the surgical site and no residual lymphatic malformation. Indication for surgical removal instead of ultrasound-guided sclerosis was decided, based on the large size of the malformation and its close position to main vessels. We clarified these aspects in the Results section, according to the Reviewer’s questions.

c. We edited Figure 3, by deleting image b, according to the Reviewer’s suggestion.

d. Indication for surgery in patients with thyroglossal duct cysts was rapid increase in size (n=4) and infection (n=3). Infected cysts were operated after resolution of the acute episode. We reported this information in the Results section, according to the Reviewer’s suggestion.

e. Following the Reviewer’s comment, we edited the Data Availability Statement as follows: “Data supporting the findings of this study are available from the corresponding author C.E. on request.”.